# The Effects of Sodium Butyrate, Coated Sodium Butyrate, and Butyric Acid Glycerides on Nutrient Digestibility, Gastrointestinal Function, and Fecal Microbiota in Turkeys

**DOI:** 10.3390/ani12141836

**Published:** 2022-07-19

**Authors:** Zbigniew Makowski, Krzysztof Lipiński, Magdalena Mazur-Kuśnirek

**Affiliations:** Department of Animal Nutrition and Feed Science, University of Warmia and Mazury in Olsztyn, 10-718 Olsztyn, Poland; zbyszek-tu@o2.pl (Z.M.); krzysztof.lipinski@uwm.edu.pl (K.L.)

**Keywords:** butyric acid glycerides, coated sodium butyrate, digestibility, fecal bacteria, gastrointestinal parameters, organic acids, poultry, sodium butyrate

## Abstract

**Simple Summary:**

The development of antimicrobial resistance is one the most serious health threats. Therefore, alternatives to antibiotic growth promoters that could be used in animal production are sorely needed. Organic acids, including butyric acid, are among the most promising compounds. Butyrate exhibits antimicrobial activity; it decreases the pH of intestinal digesta and decreases the amounts of pathogenic microbes. In this study, turkeys were fed diets with various forms of butyric acid. The growth performance of turkeys and duodenal villus height increased, whereas the fecal populations of *Escherichia coli* and *Clostridium perfringens* decreased when butyric acid glycerides or coated sodium butyrate were added to turkey diets. An increase in protein digestibility and a decrease in the gizzard pH were noted in birds fed diets with butyric acid glycerides. The addition of butyric acid in different forms to diets increased the butyric acid concentration in the cecal digesta of turkeys. The study results suggest that protected forms of butyric acid improve growth performance and protein digestibility and decrease the fecal populations of pathogenic bacteria in turkeys. Therefore, protected forms of butyric acid can be valuable feed additives in turkey nutrition.

**Abstract:**

This study aimed to determine the efficacy of sodium butyrate (SB), coated sodium butyrate (CSB), and butyric acid glycerides (BAG) in turkey nutrition based on an analysis of nutrient digestibility, gastrointestinal function, and fecal bacterial populations. A total of 400 1-day-old female BIG 6 turkeys were divided into 4 groups, with 5 replicates per group and 20 birds per replicate, to determine the effects exerted by various forms of butyric acid (SB, CSB, and BAG). The addition of CSB and BAG to turkey diets improved the feed conversion ratio (FCR, *p* < 0.05), increased the values of the European Efficiency Index (EEI, *p* < 0.01) and duodenal villus height (*p* < 0.05), and decreased the fecal populations of *Escherichia coli* and *Clostridium perfringens* (*p* < 0.05). Dietary supplementation with BAG increased protein digestibility (*p* < 0.05). The analyzed forms of butyrate added to turkey diets increased the butyric acid concentration in the cecal digesta (*p* < 0.01). The results of this study indicate that protected forms of butyric acid can be valuable feed additives in turkey nutrition.

## 1. Introduction

In poultry production, dietary treatments and modifications should promote intestinal homeostasis. Organic acids have been reported as antibacterial, immune-potentiating, and growth promoters in poultry [1]. The ability of acids to reduce the gut pH and their antimicrobial effects depend on their dissociation status and *p*KA value [2]. The acid dissociation constant (*p*Ka), i.e., the pH at which half of the acid has dissociated, is one of the key characteristics of these compounds. Organic acids with higher *p*Ka values are more effective antimicrobial compounds [3]. A positive influence of different organic acids (including formic, fumaric, lactic, and citric acids) on the performance and health status of birds has been well documented [4,5,6,7].

Butyric acid (BA) is produced by microbial fermentation, mostly in the colon. It is the main source of energy for colonocytes, and it directly affects the proliferation, maturation, and differentiation of mucosal cells [8,9]. Thus, BA can influence gene expression and protein synthesis [10], and improve protein digestibility [11,12]. Butyrate contributes to enhancing immunity, decreases the pH in the gastrointestinal tract (GIT), and suppresses the growth of pathogenic bacteria [13,14,15,16].

The derivatives of BA include salts, and sodium butyrate (SB) is the most commonly used form [17]. Butyric acid is characterized by a *p*KA (4.81) that dissociates in the crop. Protected forms of BA, such as coated SB (CSB) and BA glycerides (BAG) obtained by esterification of BA with glyceride, are available in higher concentrations in the distal segments of the small intestine [12,18].

Recent studies have demonstrated that supplementation of broiler chicken diets with BA improved the performance of birds and nutrient digestion and absorption, and reduced disease incidence [5,19,20,21]. However, there are no published studies investigating the effects of various forms of butyric acid on the nutrient digestibility and gastrointestinal tract of turkeys.

The research hypothesis postulates that the supplementation of turkey diets with different forms of BA can improve nutrient digestibility and gastrointestinal function, and reduce the size of fecal bacterial populations, and that the efficacy of butyrate depends on its form (unprotected salts, coated salts, glycerides). This study aimed to determine the efficacy of SB, CSB, and BAG in turkey nutrition based on an analysis of the nutrient digestibility, gastrointestinal function, and fecal microflora populations.

## 2. Materials and Methods

### 2.1. Animals and Diets

A total of 400 1-day-old female BIG 6 turkeys were divided into 4 groups, with 5 replicates per group and 20 birds per replicate. The experiment lasted for 15 weeks. The turkeys were raised on a litter in pens, under standard housing conditions. The animals had free access to drinking water and feed. Mashed diets were formulated based in wheat-corn-soybean (Table 1) and formulated according to the. NRC nutritional recommendations [22]. The feed samples were analyzed for their content of dry matter (DM), ether extract (EE), crude fiber (CF), crude protein (CP), and crude ash, according to AOAC [23]. The metabolizable energy (AME_N_) and the concentrations of lysine (Lys), methionine (Met), cysteine (Cys), and minerals in the feeds were calculated according to the Nutrient Requirements of Poultry [22].

The control group (I) received a basal diet without any additives. In group II, turkeys were administered sodium butyrate (concentration of n–butyric acid 98 ± 2%; Adimix CPS^®^; Nutriad) in a dose of 1 kg/t of feed. In group III, birds’ diets were supplemented with 3.3 kg/t coated sodium butyrate (concentration coated n–butyric acid 30 ± 2%; Adimix 30 Coated^®^; Nutriad). In experimental group IV, butyric acid glyceride (BAG, concentration of n-butyric acid 23–26%; C4 powder Monobutyrin^®^; Silo S.r.l.—monoglycerides, diglycerides, triglycerides of butyric acid) was added to turkey diets in a dose of 3.4 kg/t of feed.

The body weight (BW) of turkeys, feed intake, and mortality were measured during the experimental period. The feed conversion ratio (FCR) was calculated by dividing the feed intake by the body weight gain for each pen. The European Efficiency Index (EEI) was calculated according to the following formula:EEI = the livability (%) × BW (kg)/the age (days) × FCR (kg) × 100

### 2.2. Sample Collection and Laboratory Analyses

The ileal digestibility of nutrients (DM, CP, and EE) was analyzed with the use of 0.3% chromium oxide (Cr_2_O_3_) as an indigestible marker in weeks 6 and 15 of the experiment. The marker was added to the diets during the last week of each experimental period, on four consecutive days. Next, eight birds per pen were slaughtered at 6 and 15 weeks of age. Digesta were collected by flushing the distal two-third segment of the intestine, between the Meckel’s diverticulum and 2 cm anterior to the ileo-ceca-colonic junction, using distilled water [24]. The intestinal contents were instantly frozen at 18 °C, freeze-dried, and ground through a 0.5-mm screen for later chemical analyses. Dried digesta and feed samples were analyzed for their content of DM, CP, and EE according to AOAC procedures [23]. The content of Cr_2_O_3_ was determined in the diets and digesta with the use of an atomic absorption spectrophotometer (Shimadzu, AA 670, Tokyo, Japan), as described by Williams et al. [25]. The digestibility of nutrients was based on the following formula:Digestibility (%) = 100 − 100 × [(Cr_2_O_3_ Diet × DM, CP, EE Digesta) / (Cr_2_O_3_ Digesta × DM, CP, EE Diet)]
in which Cr_2_O_3_ Diet and Cr_2_O_3_ Digesta represent the Cr_2_O_3_ concentration in the feed and digesta samples (g/kg), respectively; and DM, CP, EE Diet and DM, CP, EE Digesta stand for the concentrations of DM, CP, and EE in the feed and digesta samples (g/kg), respectively.

At 6 and 15 weeks of age, 60 birds (3 from each replicate) were killed via cervical dislocation to analyze the GIT structure and function. Segments of the GIT (crop, proventriculus, gizzard, small intestine, and ceca) were weighed with and without the contents. The total length of the small intestine and ceca was measured.

In the digesta of the crop, gizzard, proventriculus, small intestine, and ceca, the pH value was determined using a pH meter. The small intestinal contents were collected for viscosity evaluation. The fresh digesta was diluted with 1:1 deionized water and centrifuged at 5200 g for 20 min (centrifuge MPW-350 R; rotor No. 12,436). The supernatant was withdrawn and determined at 39 °C using a Fungilab rotational viscometer (v. 1.2 Alpha Series 101,427, Barcelona, Spain).

Histomorphological analysis included determination of the villi height and crypt depth. Duodenal samples were fixed in neutral-buffered formalin (10%) at pH 7.4. The microtome sections (4 μm thick) were stained with hematoxylin and eosin. The morphological characteristics were determined using an optical microscope (Olympus BX-50, Warsaw, Poland) and a digital camera connected to a personal computer with Cell^B (OLYMPUS) software.

The concentrations of short-chain fatty acids (SCFAs) in the cecum of turkeys were analyzed by gas chromatography (Shimadzu GC 14A, Shimadzu Co., Kyoto, Japan).

In weeks 6 and 15 of the experiment, excreta samples were collected from 10 turkeys in each group for microbiota analyses, which were performed immediately after sampling. *Lactobacilli* strains were verified by enumeration of the MRS agar (pH 6.4 ± 0.1) using a double-layer technique, incubated in 37 °C for 72 h, and further confirmed by morphological cell observation. The most probable number (MPN) of *E. coli* was determined using brilliant green bile broth, after incubation at 37 °C for 48 h, subsequent transfer of positive samples into tryptone water, followed by overnight incubation at 44 °C, and the indole test [26]. *Clostridium* spp. samples were prepared in anaerobic conditions and were cultured in an anaerobic jar using the Anaerocult technique (MERCK). Tryptose Sulfite Cycloserine (TSC) was used for the enumeration of *C. perfringens*.

### 2.3. Statistical Analysis

One-way analysis of variance (ANOVA) was performed with the use of Statistica 12.0 software (StatSoft. Kraków, Poland). When a significant treatment effect was noted, the post-hoc Duncan test was used to determine differences between treatment groups. The arithmetic mean, standard error of the mean (SEM), and level of significance (*p* < 0.05) were calculated for all results.

## 3. Results

### 3.1. Growth Performance

No significant differences in final BW or livability were observed between turkeys fed the control, SB, CSB, and BAG diets. (Table 2). Birds that received diets with the coated sodium butyrate or butyric acid glycerides were characterized by lower values of the FCR (*p* < 0.05) as compared with the control group. Turkeys whose diets were administered with coated sodium butyrate or butyric acid glycerides had higher values of the EEI (*p* < 0.01) than animals from the other groups.

### 3.2. Nutrient Digestibility

In week 6, protein digestibility improved (*p* < 0.05) in turkeys fed BAG-supplemented diets compared with animals from the control group (Table 3). The tested additives had no influence on DM or EE digestibility in weeks 6 and 15.

### 3.3. Gastrointestinal Tract

Turkeys fed BAG-supplemented diets had a lower (*p* < 0.05) pH of the gizzard digesta than those fed without additive or SB-supplemented diets (Table 4). Birds fed BAG-supplemented diets had a shorter (*p* < 0.05) small intestine than those fed SB-supplemented diets. An increase in the height of the intestinal villi was noted in turkeys administered coated sodium butyrate or butyric acid glycerides compared with control group birds.

Birds fed BAG-supplemented diets had a higher (*p* < 0.05) pH of the crop digesta than those from remaining groups (Table 5). The gizzard weight was lower (*p* < 0.05) in turkeys that received diets supplemented with sodium butyrate and the coated form of butyrate than in birds from the remaining groups. The tested additives had no effect on the small intestinal parameters. Turkeys administered protected forms of butyric acid were characterized by a decreased cecal weight, relative to the other groups.

### 3.4. Concentrations of SCFAs in the Cecal Digesta and Fecal Microbiota

The concentrations of SCFAs are presented in Table 6 and Table 7. In week 6, the analyzed feed additives had no influence on the SCFAs concentrations in the cecal digesta. In week 15, the butyric acid concentration in the cecal digesta was higher in turkeys fed different forms of butyric acid, compared with control group birds (*p* < 0.01). Birds fed SB- or CSB-supplemented diets were characterized by a lower (*p* < 0.05) concentration of propionic acid (%) in the cecal digesta of the turkeys, relative to birds fed the control diets.

In week 6, the population size of *C. perfringens* decreased in fecal samples collected from birds that received protected forms of butyric acid, compared with control group birds (*p* < 0.05) (Table 8). In week 15, the inclusion of coated sodium butyrate and butyric acid glycerides in the turkey diets contributed to a decrease in the counts of *E. coli* and *C. perfringens* in the fecal samples (*p* < 0.05). The fecal population of *Lactobacillus* was not affected by the tested additives.

## 4. Discussion

The maintenance of the gastrointestinal tract is vital for poultry productivity. The small intestine and intestinal villi are the main site for digestion and absorption of nutrients [27]. A better villus height increases the surface area for nutrient absorption, thus enhancing nutrient digestion and absorption and, in consequence, improving the growth performance of birds [28]. Butyric acid stimulates the growth of intestinal villi [29,30], which was also observed in this study. The height of intestinal villi increased significantly in turkeys fed diets supplemented with protected forms of BA. Additionally, lower pH levels in the gizzard stimulate the activity of endogenous enzymes that improve the nutrient digestibility, thus enhancing bird performance [16,31]. Our current study’s found greater protein digestibility and lower pH in the gizzard of birds fed diets with BAG. When butyric acid is given to poultry, it is immediately metabolized and absorbed in the crop and in the acidic conditions of the stomach [21]. Encapsulation of butyric acid positively affects digestive processes and facilitates the release of active substances in the distal segments of the GIT [21]. As a result, enterocytes can utilize butyrate as a source of energy [32,33]. This results in a higher concentration of BA in the duodenum and leads to higher proteolytic activity in birds [34]. The observed changes can be attributed to an increased intestinal absorptive surface area and lower pH value, contributing to more efficient digestion and absorption [35,36]. Other authors have also noted an increase in the height of jejunal villi [21,37,38,39] and the villus height/crypt depth ratio in birds fed diets with the addition of encapsulated butyrate [21,37].

The poultry microbiome consists of both beneficial bacteria (e.g., Gram-positive *lactobacilli* and *bifidobacteria*) and pathogenic bacteria (e.g., *E. coli* and *Clostridium* spp.) [28]. The addition of BA to poultry diets can have a positive influence on the health status of birds by reducing the counts of harmful microbes. In birds, butyrate stimulates the secretion of mucins that have antimicrobial properties and suppress the growth of pathogenic bacteria such as *Clostridium* spp., *Salmonella* spp., and *E. coli* [40]. There are no published studies investigating the effect of different forms of butyric acid on the fecal microbiome. In our study, a significant increase in the BA concentration and a decrease in the proportion of propionic acid in the cecal digesta were noted in 15-week-old turkeys fed diets supplemented with different forms of BA. SCFAs play an important role in maintaining the structural and functional integrity of intestinal epithelial cells [41]. One of the mechanisms by which the fecal microflora may reduce harmful microbe is the bacteriostatic effect of volatile fatty acids, including butyric acid in the ceca [42].

Butyric acid is rapidly absorbed in the foregut when it is in a mixture in the form of salt [43]. However only undissociated forms such as capsulated or glycerides reach the lower part of GIT to exert a stronger bacteriostatic effect [42], which was confirmed in the present study. In the current study, the concentrations of pathogenic bacteria (*E. coli* and *C. perfringens*) decreased in the fecal samples collected from turkeys fed diets supplemented with protected forms of BA. The undissociated form of butyric acid can dissociate into butyrate and release H^+^ ions inside the pathogenic bacteria cytoplasm [40,44]. This lowers the pH value in the gizzard and intestine of birds, causing inhibition of pathogenic bacteria’s growth [45]. Moreover, harmful microbes use up energy by activating proton pumps to struggle with the lowering of the acid in the cell wall, thus disturbing pathogens’ colonization in poultry intestines [17], whereas BA’s antibacterial activity creates an acidic environment in the stomach (pH 3.5–4.0), which prevents the growth of other pathogenic Gram-negative bacteria in the GIT of animals [14,29].

In the present study, the improvement in selected performance parameters such as the FCR and EEI in turkeys that received protected forms of BA can be partly explained by the improved protein digestibility due to an increased intestinal absorptive surface area. A positive effect of CSB and BAG on the feed conversion ratio was also reported by other authors [21,37,39,46].

## 5. Conclusions

Coated butyric acid or butyric acid glyceride acid supplementation in turkey diets showed positive effects on feed conversion, intestinal morphology, and health of birds by lowering the pathogenic concentration in the feces. The protected forms of butyric acid can be valuable feed additives in turkey nutrition.

## Figures and Tables

**Table 1 animals-12-01836-t001:** Composition of the basal diets and the calculated nutritional value.

Specification	Starter 1	Starter 2	Grower 1	Grower 2	Finisher
	0–3 Weeks	4–6 Weeks	7–9 Weeks	10–12 Weeks	13–15 Weeks
Ingredient [g/kg]					
Wheat	261.9	310.3	416.7	515.8	590.4
Corn	200.0	200.0	150.0	100.0	100.0
Soybean meal	358.2	360.9	347.4	300.0	225.1
Full-fat soybeans	100.0	50.0	-	-	-
Blood meal	20.0	10.0	-	-	-
Soybean oil	5.2	19.2	39.1	45.1	47.8
L-lysine HCl	3.1	3.6	3.7	3.2	4.0
DL-methionine	3.5	2.6	2.4	2.6	2.5
L-threonine	0.7	0.7	1.1	0.7	1.0
Limestone	18.8	14.5	13.4	11.1	9.7
Calcium phosphate	22.1	19.9	17.7	13.1	11.1
Sodium bicarbonate	0.1	1.3	1.2	0.7	0.7
NaCl	2.0	1.9	2.2	2.6	2.7
Feed enzymes ^1^	0.1	0.1	0.1	0.1	0.1
Premix *	5.0	5.0	5.0	5.0	5.0
Nutritional value					
AME_N_, [kcal/kg]	2800	2880	3000	3100	3200
CP, %	27.51	25.49	23.24	22.31	20.13
CF, %	3.22	2.96	2.90	3.19	3.22
EE, %	3.52	4.13	5.31	5.77	6.66
Lys, [%]	1.77	1.65	1.45	1.30	1.17
Met + Cys, [%]	1.15	1.02	0.95	0.93	0.85
Ca, [g]	1.40	1.20	1.15	1.00	0.90
P available., [g]	0.70	0.65	0.60	0.50	0.45
Na, [g]	0.13	0.15	0.15	0.15	0.15

^1^—Xylanase, phytase; * Premix composition: Starter—12,500 IU vitamin A, 4500 IU vitamin D_3_, 87.5 mg vitamin E, 3.75 mg vitamin K_3_, 3.5 mg vitamin B_1_, 10 mg vitamin B_2_, 75 mg niacin, 22.5 mg pantothenic acid, 6.0 mg vitamin B_6_, 30 µg vitamin B_12_, 2.5 mg folic acid, 400 µg biotin, 800 mg choline chloride, 92.5 mg Fe, 130 mg Mn, 20 mg Cu, 105 mg Zn, 2.5 mg I, 0.3 mg Se; Grower—11,500 IU vitamin A, 4140 IU vitamin D_3_, 80.5 mg vitamin E, 3.45 mg vitamin K_3_, 3.22 mg vitamin B_1_, 9.2 mg vitamin B_2_, 69 mg niacin, 20.7 mg pantothenic acid, 5.52 mg vitamin B_6_, 37.6 µg vitamin B_12_, 2.3 mg folic acid, 368 µg biotin, 600 mg choline chloride, 85.1 mg Fe, 120 mg Mn, 18.4 mg Cu, 96.6 mg Zn, 2.3 mg I, 0.26 mg Se; Finisher—10,500 IU vitamin A, 3780 IU vitamin D_3_, 66.5 mg vitamin E, 2.85 mg vitamin K_3_, 2.66 mg vitamin B_1_, 7.6 mg vitamin B_2_, 57 mg niacin, 17.1 mg pantothenic acid, 4.6 mg vitamin B_6_, 22.8 µg vitamin B_12_, 1.9 mg folic acid, 304 µg biotin, 400 mg choline chloride, 70.3 mg Fe, 98.8 mg Mn, 15.2 mg Cu, 79.8 mg Zn, 1.9 mg I, 0.23 mg Se.

**Table 2 animals-12-01836-t002:** Effect of the dietary treatments on the performance of turkeys.

Specification	Groups				SEM	*p*
	I-Control	II-SB	III-CSB	IV-BAG		
Duration, days	105	105	105	105	-	-
Final BW, g	9.59	9.67	9.78	9.79	0.033	0.087
FCR, kg/kg	2.35 ^a^	2.32 ^ab^	2.29 ^b^	2.28 ^b^	0.008	0.014
Mortality, %	7.00	7.00	7.00	6.00	0.547	0.907
EEI, points	407.65 ^a^	417.19 ^a^	427.75 ^b^	429.87 ^b^	2.537	<0.001

Different superscript lower-case letters a and b indicate a significant difference at *p* < 0.05; SEM = standard error of the mean; FCR—feed conversion ratio; EEI—European Efficiency Index; II-SB—sodium butyrate, 1 kg/t; III-CSB—coated sodium butyrate, 3.3 kg/t; IV-BAG—butyric acid glycerides, 3.4 kg/t.

**Table 3 animals-12-01836-t003:** Effect of butyric acid supplementation on the ileal nutrient digestibility in turkeys.

	Groups					
Item	I-Control	II-SB	III-CSB	IV-BAG	SEM	*p*
6th week						
DM	75.88	76.56	77.08	77.31	0.348	0.491
CP	81.02 ^a^	82.33 ^ab^	82.52 ^ab^	83.22 ^b^	0.274	0.030
EE	89.61	90.61	90.81	90.57	0.351	0.641
15th week						
DM	80.44	80.84	81.34	81.47	0.282	0.459
CP	83.94	84.64	84.84	84.91	0.301	0.669
EE	93.52	93.32	93.42	93.41	0.321	0.997

Different superscript lower-case letters a and b indicate a significant difference at *p* < 0.05; SEM = standard error of the mean; II-SB—sodium butyrate, 1 kg/t; III-CSB—coated sodium butyrate, 3.3 kg/t; IV-BAG—butyric acid glycerides, 3.4 kg/t.

**Table 4 animals-12-01836-t004:** Gastrointestinal parameters in 6-week-old turkeys.

	Groups					
Item	I-Control	II-SB	III-CSB	IV-BAG	SEM	*p*
Crop						
weight, g/kg of BW	3.58	3.80	3.70	3.98	0.095	0.502
pH of digesta	5.08	4.84	4.93	4.92	0.071	0.703
Proventriculus						
weight, g/kg of BW	3.45	3.59	3.72	3.50	0.047	0.202
pH of digesta	3.79	3.67	3.68	3.74	0.069	0.935
Gizzard						
weight, g/kg of BW	16.05	15.63	16.27	15.66	0.217	0.683
pH of digesta	3.90 ^ab^	4.03 ^a^	3.81 ^abc^	3.62 ^c^	0.048	0.016
Small intestine						
length, cm/kg BW	178.60	182.40	173.20	172.10	1.703	0.108
weight, g/kg of BW	23.46	24.19	22.76	22.02	0.350	0.149
pH of digesta	6.27	6.14	6.10	6.10	0.070	0.834
viscosity, mPas.s	2.39	2.16	2.26	2.28	0.063	0.691
villus height, µm	1258,01 ^a^	1498.85 ^ab^	1560.00 ^b^	1630.89 ^b^	48.747	0.035
crypt depth, µm	251.83	249.27	234.24	269.03	6.442	0.307
Caeca						
length, cm/kg BW	48.65	48.15	49.20	50.00	0.713	0.830
weight, g/kg of BW	6.91	6.94	6.39	7.07	0.152	0.415
pH of digesta	5.61	5.33	5.48	5.35	0.057	0.282

Different superscript lower-case letters a, b, and c indicate a significant difference at *p* < 0.05; II-SB—sodium butyrate, 1 kg/t; III-CSB—coated sodium butyrate, 3.3 kg/t; IV-BAG—butyric acid glycerides, 3.4 kg/t.

**Table 5 animals-12-01836-t005:** Gastrointestinal parameters in 15-week-old turkeys.

	Groups					
Item	I-Control	II-SB	III-CSB	IV-BAG	SEM	*p*
Crop						
weight, g/kg of BW	2.95	3.22	3.18	2.74	0.114	0.433
pH of digesta	4.58 ^b^	4.52 ^b^	4.66 ^b^	4.96 ^a^	0.054	0.016
Proventriculus						
weight, g/kg of BW	1.35	1.45	1.37	1.34	0.031	0.559
pH of digesta	3.50	3.48	3.53	3.29	0.087	0.779
Gizzard						
weight, g/kg of BW	9.34 ^a^	8.13 ^b^	8.08 ^b^	8.71 ^a^	0.168	0.020
pH of digesta	3.92	3.67	4.05	3.82	0.063	0.183
Small intestine						
length, cm/kg BW	241.00	236.20	234.60	227.80	2.941	0.469
weight, g/kg of BW	10.64	10.71	10.17	10.72	0.170	0.634
pH of digesta	6.13	5.89	5.99	6.15	0.081	0.666
viscosity, mPas.s	2.59	2.46	2.46	2.52	0.063	0.618
villus height, µm	2371.77	2666.28	2635.41	2611.77	46.947	0.098
crypt depth, µm	225.93	247.07	250.72	257.44	5.393	0.190
Caeca						
length, cm/kg BW	68.60	73.75	73.15	69.00	1.086	0.202
weight, g/kg of BW	3.55 ^a^	3.56 ^a^	3.08 ^b^	3.31 ^b^	0.070	0.044
pH of digesta	5.16	5.16	5.57	5.48	0.070	0.067

Different superscript lower-case letters a and b indicate a significant difference at *p* < 0.05. SEM = standard error of the mean; II-SB—sodium butyrate, 1 kg/t; III-CSB—coated sodium butyrate, 3.3 kg/t; IV-BAG—butyric acid glycerides, 3.4 kg/t.

**Table 6 animals-12-01836-t006:** Concentrations of short-chain fatty acids (µmol/g *) in the cecal digesta of 6-week-old turkeys.

	Groups					
Item	I-Control	II-SB	III-CSB	IV-BAG	SEM	*p*
Total SCFAs	4.99	6.69	5.31	8.57	0.587	0.118
Acetic acid	1.36	2.44	1.65	3.34	0.302	0.122
Propionic acid	0.13	0.05	0.09	0.12	0.020	0.328
Isobutyric acid	0.08	0.29	0.14	0.34	0.048	0.244
Butyric acid	0.71	1.18	1.04	1.85	0.157	0.098
Valeric acid	0.03	0.0	0.03	0.0	0.006	0.126
C_2_ profile, %	30.69	38.65	31.64	37.44	2.320	0.541
C_3_ profile, %	28.07	34.72	37.96	30.68	4.141	0.855
C_4_ profile, %	27.72	25.70	25.41	23.82	2.570	0.967

SEM = standard error of the mean; II-SB—sodium butyrate, 1 kg/t; III-CSB—coated sodium butyrate, 3.3 kg/t; IV-BAG—butyric acid glycerides, 3.4 kg/t; *—fresh digesta.

**Table 7 animals-12-01836-t007:** Concentrations of short-chain fatty acids (µmol/g *) in the cecal digesta of 15-week-old turkeys.

	Groups					
Item	I-Control	II-SB	III-CSB	IV-BAG	SEM	*p*
Total SCFAs	8.80	10.73	11.02	14.27	0.769	0.084
Acetic acid	1.98	2.53	2.95	3.23	0.249	0.318
Propionic acid	1.53	1.82	1.74	1.86	0.159	0.894
Isobutyric acid	0.06	0.15	0.08	0.03	0.024	0.352
Butyric acid	1.87 ^a^	3.60 ^b^	3.00 ^b^	3.88 ^b^	0.215	0.001
Valeric acid	0.45	0.80	0.64	0.74	0.122	0.771
C_2_ profile, %	21.49	22.19	25.34	22.92	1.394	0.827
C_3_ profile, %	48.01 ^a^	31.70 ^b^	32.59 ^b^	40.86 ^ab^	2.417	0.045
C_4_ profile, %	24.22	37.26	31.58	28.67	1.879	0.084

Different superscript lower-case letters a and b indicate a significant difference at *p* < 0.05; II-SB—sodium butyrate, 1 kg/t; III-CSB—coated sodium butyrate, 3.3 kg/t; IV-BAG—butyric acid glycerides, 3.4 kg/t; *—fresh digesta.

**Table 8 animals-12-01836-t008:** Effect of butyric acid on the fecal microbiota (log_10_ CFU/g) in turkeys.

	Groups					
Item	I-Control	II-SB	III-CSB	IV-BAG	SEM	*p*
6th week						
*Lactobacillus* spp.	7.93	8.11	8.20	8.26	0.046	0.067
*Escherichia coli*	6.04	6.01	5.93	5.89	0.068	0.871
*Clostridium perfringens*	4.48 ^a^	4.03 ^ab^	3.92 ^b^	3.75 ^b^	0.086	0.012
15th week						
*Lactobacillus* spp.	7.39	7.54	7.43	7.63	0.042	0.163
*Escherichia coli*	6.46 ^a^	6.42 ^a^	6.10 ^b^	6.11 ^b^	0.052	0.012
*Clostridium perfringens*	4.22 ^a^	3.83 ^ab^	3.69 ^b^	3.62 ^b^	0.082	0.039

Different superscript lower-case letters a and b indicate a significant difference at *p* < 0.05; II-SB—sodium butyrate, 1 kg/t; III-CSB—coated sodium butyrate, 3.3 kg/t; IV-BAG—butyric acid glycerides, 3.4 kg/t.

## Data Availability

The datasets generated during and/or analyzed during the current study are available from the corresponding author on reasonable request.

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
