# Peer review of "The Effects of Sodium Butyrate, Coated Sodium Butyrate, and Butyric Acid Glycerides on Nutrient Digestibility, Gastrointestinal Function, and Fecal Microbiota in Turkeys"

_animals, 2022, doi:10.3390/ani12141836_

Round 1

Reviewer 1 Report

Thank you for the opportunity to review this interesting and well-prepared manuscript.

A few comments and suggestions:

In the simple summary and abstract, specify that villus height was measured in the duodenum (i.e. dudenal villi height).

Introduction, line 55: Butyric acid is characterized...

Materials and Methods:

Line 73 and elsewhere in the manuscript: Generally, numbers less than 10 are written out (four groups, five replicates).

Line 80, 81: Amino acids not in uppercase (lysine, methionine, cysteine).

Table 1: Feed enzymes should be specified. This may have important implications for the digestibility results and should also be discussed in the Discussion section. 

Table 1, Premix composition: A letter "J" was used for an element in all three phases. Is this supposed to be "I" perhaps, for iodine?

Line107, 108: Surely it should read, 'FCR was calculated by dividing feed intake by body weight gain for each pen' and not the other way around as stated currently? 

Line 116: State method of euthanasia (the method of euthanasia may affect digestibility results as the excessive contractions with cervical dislocation may cause movement of the digesta).

Line 119 and elsewhere: Use the correct symbol for degrees celsius.

Line 136: '...pH meter. Small intestine... (insert .)

Line 145: personal computer, not PC.

Results

Line 169, 174, 183 and elsewhere: Please use the journal's standard/accepted format for p<0.05 (spacing and P or p).

Discussion:

The Discussion section can be improved. Currently, the Discussion section consists mainly of a repetition of results and comparison with existing literature, while little attempt is made to explain the mechanisms of actions. 

Line 298: 'SCFAs affect on maintaining...' check grammar, sentence construction. Also, this statement should be supported by references.

Author Response

Thank you for the opportunity to review this interesting and well-prepared manuscript.

We would like to thank the Reviewer for a thorough perusal of the manuscript and valuable comments and suggestions which have enabled us to improve its quality.

A few comments and suggestions:

In the simple summary and abstract, specify that villus height was measured in the duodenum (i.e. dudenal villi height).

The information was added. L 15 and 29

Introduction, line 55: Butyric acid is characterized...

The relevant correction has been made L 72

Materials and Methods:

Line 73 and elsewhere in the manuscript: Generally, numbers less than 10 are written out (four groups, five replicates).

The relevant correction has been made L 90

Line 80, 81: Amino acids not in uppercase (lysine, methionine, cysteine).

The relevant correction has been made L 97

Table 1: Feed enzymes should be specified. This may have important implications for the digestibility results and should also be discussed in the Discussion section. 

Feed enzymes added to the mixture were xylanase and phytase. The missing information was added under Table 1

 These days all broiler or turkey feed contains enzymes such as xylanase or phytase that breakdown non-starch polysaccharides.  In our study, birds from all groups fed diets with these enzymes, so the differences in digestibility of nutrients between the groups we try explain in rewritten discussion section L. 364-366

Table 1, Premix composition: A letter "J" was used for an element in all three phases. Is this supposed to be "I" perhaps, for iodine?

The relevant correction has been made under Table 1

Line107, 108: Surely it should read, 'FCR was calculated by dividing feed intake by body weight gain for each pen' and not the other way around as stated currently? 

The relevant correction has been made L  137

Line 116: State method of euthanasia (the method of euthanasia may affect digestibility results as the excessive contractions with cervical dislocation may cause movement of the digesta).

The birds from each treatment were killed via cervical dislocation. This method of killing birds is recommended by Local Ethics Committee for Animal Experimentation in Olsztyn, Poland. This method of euthanasia is also used by other researchers Kaczmarek et al., (2016).

Kaczmarek, S.A.; Barri, A.; Hejdysz, M.; Rutkowski, A. Effect of Different Doses of Coated Butyric Acid on Growth Performance and Energy Utilization in Broilers. Poult. Sci. 2016, 95, 851–859.

Line 119 and elsewhere: Use the correct symbol for degrees celsius.

Line 136: '...pH meter. Small intestine... (insert .)

Line 145: personal computer, not PC.

The relevant corrections have been made L 156, 173, 182, 191

Results

Line 169, 174, 183 and elsewhere: Please use the journal's standard/accepted format for p<0.05 (spacing and P or p).

The relevant corrections have been made  through the manuscript.

Discussion:

The Discussion section can be improved. Currently, the Discussion section consists mainly of a repetition of results and comparison with existing literature, while little attempt is made to explain the mechanisms of actions. 

The discussion was rewritten.

Line 298: 'SCFAs affect on maintaining...' check grammar, sentence construction. Also, this statement should be supported by references.

The relevant corrections have been made L391

Reviewer 2 Report

Please add "calculated nutritional value" instead of just "nutritional value" to table 1

Please explain why increases in dig of CP but not in EE nor DM? (Table 3)

Author Response

We would like to thank the Reviewer for a thorough perusal of the manuscript and valuable comments and suggestions which have enabled us to improve its quality.

Please add "calculated nutritional value" instead of just "nutritional value" to table 1

 The relevant correction has been made L101

Please explain why increases in dig of CP but not in EE nor DM? (Table 3)

The EE and DM digestibility was also higher in birds fed butyric addition diets in compare with control group. However the differences wasn’t statistically significant.

We rewrite the section “Discussion” and we took attempt to explain many mechanism, including digestion L. 364-366

Reviewer 3 Report

   I read the manuscript entitled “The effects of sodium butyrate, coated sodium butyrate and butyric acid glycerides on nutrient digestibility, gastrointestinal function and fecal microbiota in turkeys” for possible publication in Animals. The experimental design, methods and conducting are appropriate, but the statistical analyses were not done enough and, therefore, the summarized results are not clear. I recommend that the manuscript should be revised for the following points:

1)    The statistical analyses were not done enough. The authors used Duncan test for multiple comparisons of the means in treatment groups. The method is considered unsuitable in this study for two reasons. First, Duncan test will not perform well with unequal sample sizes. Second, Duncan test does not control experimentwise error rate. The authors should carry out multiple comparisons of the means by other multiple range test, such as tukey’s test.

2)    The formatting of the reference list in L356-483 should be applied to the Animals format. For example, “Microencapsulated Short-Chain Fatty Acids in Feed Modify Colonization and Invasion Early after Infection with Salmonella Enteritidis in Young Chickens. Poult. Sci. 2004, 83, 69–74.” in L358-359 should be changed to “Microencapsulated short-chain fatty acids in feed modify colonization and invasion early after infection with Salmonella enteritidis in young chickens. Poult. Sci. 2004, 83, 69–74.”.  In addition, the accumulate information in each article should be noted. For example, “479485” (article No) should be added in L368; “589” in L372 should be changed to “589-594”; “541” in L405 should be changed to “541-551”; “390” in L432 should be changed to “390-396”; “2329-2334” should be added in L480.

3)    Others: “(AMEN)” in L80 should be changed to “(AMEN)”; “for was calculated” in L107 should be deleted; Put a period after “a PH meter” in L136; “37°C/48 h” in L153 should be changed to 37°C for 48 hour”; Is the sentence of “Gram-negative bacteria, including E. coli, produce acetate and propionate, whereas Gram-negative bacteria produce mainly BA [46]” in L308-309 correct? According to the article of [46], I think that gram-positive bacteria produce mainly BA. Also, the populations of Lactobacillus, one of gram-positive bacteria, are not different among the dietary treatment in this study. Therefore, I think the sentence is not required for discussion.

Author Response

We would like to thank the Reviewer for a thorough perusal of the manuscript and valuable comments and suggestions which have enabled us to improve its quality.

   I read the manuscript entitled “The effects of sodium butyrate, coated sodium butyrate and butyric acid glycerides on nutrient digestibility, gastrointestinal function and fecal microbiota in turkeys” for possible publication in Animals. The experimental design, methods and conducting are appropriate, but the statistical analyses were not done enough and, therefore, the summarized results are not clear. I recommend that the manuscript should be revised for the following points:

1)    The statistical analyses were not done enough. The authors used Duncan test for multiple comparisons of the means in treatment groups. The method is considered unsuitable in this study for two reasons. First, Duncan test will not perform well with unequal sample sizes. Second, Duncan test does not control experimentwise error rate. The authors should carry out multiple comparisons of the means by other multiple range test, such as tukey’s test.

In this research, where was multi stages and there was equal number of each group, we decided to use a one-way ANOVA analysis with Duncan test to confirm where the differences occurred between a few groups.

2)    The formatting of the reference list in L356-483 should be applied to the Animals format. For example, “Microencapsulated Short-Chain Fatty Acids in Feed Modify Colonization and Invasion Early after Infection with Salmonella Enteritidis in Young Chickens. Poult. Sci. 2004, 83, 69–74.” in L358-359 should be changed to “Microencapsulated short-chain fatty acids in feed modify colonization and invasion early after infection with Salmonella enteritidis in young chickens. Poult. Sci. 2004, 83, 69–74.”.  In addition, the accumulate information in each article should be noted. For example, “479485” (article No) should be added in L368; “589” in L372 should be changed to “589-594”; “541” in L405 should be changed to “541-551”; “390” in L432 should be changed to “390-396”; “2329-2334” should be added in L480.

The relevant corrections have been made in the references.

3)    Others: “(AMEN)” in L80 should be changed to “(AMEN)”; “for was calculated” in L107 should be deleted; Put a period after “a PH meter” in L136; “37°C/48 h” in L153 should be changed to “37°C for 48 hour”; Is the sentence of “Gram-negative bacteria, including E. coli, produce acetate and propionate, whereas Gram-negative bacteria produce mainly BA [46]” in L308-309 correct? According to the article of [46], I think that gram-positive bacteria produce mainly BA. Also, the populations of Lactobacillus, one of gram-positive bacteria, are not different among the dietary treatment in this study. Therefore, I think the sentence is not required for discussion.

 The relevant correction has been made L 97, 137, 173, 191.

The sentence ”Is the sentence of “Gram-negative bacteria, including E. coli, produce acetate and propionate, whereas Gram-negative bacteria produce mainly BA [46]” was delated.

Reviewer 4 Report

Line 35 and 36 - Keywords: use the alphabetical order

INTRODUCTION

Line 40 – when you said that “Organic acids are feed additives that act as alternative growth promoters”, it is important to emphasize that this is one of the multiple functions of organic acids. Please, do not show only this function. I believe you can delete this phrase (lines 41 and 42) or add to the next phrase and complete your information.

MATERIALS AND METHODS

Lines 74 and 75 – when you affirm that “The turkeys were raised on a litter in pens, under highly controlled environmental condition”, you are talking about totally controlled environment, not open houses, correct ? Please, the term highly is not convenient.

Lines 76 to 78 – Please, re-write this phrase: “For the nutrition of birds, standard, mash mixtures were used (Table 1) and were consistent with the nutrient requirements of turkeys.”

A suggestion: Mashed diets were formulated based in wheat-corn-soybean (Table 1) and formulated according to NRC nutritional recommendations (22).

Lines 80 to 82: In the phrase: “The metabolizable energy (AMEN), the concentration of Lysine (Lys), Methionine (Met), 80 Cysteine (Cys) and minerals in the feeds were calculated according to the Nutrient Requirements of Poultry [22].”, did you formulate using the table data or any equation proposed by NRC ?

Lines 99 to 105 – when you describe your treatments, the commercial name must be used with a symbol ® after. The product tested described as BAG is not really a glyceride compound, but a mix of butyric acid with a monoglyceride. Please, define better this product to avoid incorrect comparisons in your result description and discussion.

Lines 106 and 107: did you weigh the animals weekly ? which frequency you used ?

Line 113 – When you analyzed the ileal digestibility of nutrients (DM, CP and EE), you must be more adequate to the correct nutrition. CP is N determination, and DM and EE should be more representative if you add the energy determination in the samples. If you have this samples in your lab, I strongly recommend you determine the energy to include in this paper. Amino acid could be a very important analysis, but I understand it too expensive, but it is really a great information.

Line 128 change “where” to “in which” or simply “and”

Lines 131 and 132 – “At 6 and 15 weeks of age, sixty birds (3 from each replicate) were slaughtered to analyze GIT structure and function”, are the same birds you used for digesta collection ? Is there no interference on this procedure of collection ?

Lines 135 to 137 – “In digesta of the crop, gizzard, proventriculus, small intestine and ceca, the pH value was determined using a pH meter Small intestinal contents were collected for viscosity evaluation.” I believe there are two phrases here. Correct the punctuation, please.

Lines 146 and 147: “The concentrations of short-chain fatty acids (SCFAs) were analyzed by gas chromatography (Shimadzu GC 14A, Shimadzu Co., Kyoto, Japan).” In which fractions of the GIT ?

Lines 148 to 150. “In weeks 6 and 15 of the experiment, excreta samples were collected from 10 turkeys in each group for microbiota analyses, which were performed immediately after sampling.” Why did you not collect cecal content ?

RESULTS

In the result description, please be more precise. The use of symbols or abbreviations make more difficult the comprehension of the text. You use BA and Sb, CSB, BAG all the time.

In the first phrase “The final BW of birds fed BA-supplemented diets were similar to those of control group animals (Table 2). Birds receiving diets with the protected forms of BA were characterized by lower values of the FCR (p≤0.05) as compared with the control group. Turkeys whose diets were administered with protected forms of BA had higher values of the EEI (p≤0.01) than animals from the other groups.” It is not clear. Describe which treatment is different and compare to the active principles of the product. But use the correct description of the treatment to be more precise.

The tables 4 and 5 need to be adjusted to be correctly formatted to the page. Please, do it to clear the viewing tables in the pages

DISCUSSION

The discussion is strongly based in literature comparison, but the real discussion must be improved. The problem is that the authors tested some variables that could not be compared to other papers, like digestibility and microbiome. It is possible to note that some bacteria genera determination cannot be extrapolated to a 16-S microbiome determination.

Other phrase that was not in accordance is the last paragraph in which the authors use non-statistical differences observed in body weight to explain the results.

I recommend that the discussion should be re-written to be closer to the variables the authors tested in order to explain better the results.

CONCLUSIONS

The lines from 330 to 337 are repetition of the results. There is no conclusion, even a recommendation. Just a general information:  “The protected forms of BA can be valuable feed additives in turkey nutrition."

Author Response

We would like to thank the Reviewer for a thorough perusal of the manuscript and valuable comments and suggestions which have enabled us to improve its quality.

Line 35 and 36 - Keywords: use the alphabetical order

 The relevant correction has been made L35-36

INTRODUCTION

Line 40 – when you said that “Organic acids are feed additives that act as alternative growth promoters”, it is important to emphasize that this is one of the multiple functions of organic acids. Please, do not show only this function. I believe you can delete this phrase (lines 41 and 42) or add to the next phrase and complete your information.

 The relevant correction has been made L 40

MATERIALS AND METHODS

Lines 74 and 75 – when you affirm that “The turkeys were raised on a litter in pens, under highly controlled environmental condition”, you are talking about totally controlled environment, not open houses, correct ? Please, the term highly is not convenient.

 The relevant correction has been made L 92

The birds were kept in a building with a controlled environment. The heating and light program was in accordance with the Aviagen Turkeys (2017).

Lines 76 to 78 – Please, re-write this phrase: “For the nutrition of birds, standard, mash mixtures were used (Table 1) and were consistent with the nutrient requirements of turkeys.”

A suggestion: Mashed diets were formulated based in wheat-corn-soybean (Table 1) and formulated according to NRC nutritional recommendations (22).

The relevant correction has been made L 93

Lines 80 to 82: In the phrase: “The metabolizable energy (AMEN), the concentration of Lysine (Lys), Methionine (Met), 80 Cysteine (Cys) and minerals in the feeds were calculated according to the Nutrient Requirements of Poultry [22].”, did you formulate using the table data or any equation proposed by NRC ?

We used a table data to calculated nutritional value of each ingredients.

Lines 99 to 105 – when you describe your treatments, the commercial name must be used with a symbol ® after. The product tested described as BAG is not really a glyceride compound, but a mix of butyric acid with a monoglyceride. Please, define better this product to avoid incorrect comparisons in your result description and discussion.

According to the declaration of the producer, named of the preparation is C4 Powder Monobutyrin 25® is a mixture of monoglycerides, diglycerides, triglycerides of butyric acid and glycerol supported on silicon dioxide.

The relevant correction has been made L 130,132,133

Lines 106 and 107: did you weigh the animals weekly ? which frequency you used ?

The birds were measured weekly, but in this manuscript we decided to show only overall performance.

Line 113 – When you analyzed the ileal digestibility of nutrients (DM, CP and EE), you must be more adequate to the correct nutrition. CP is N determination, and DM and EE should be more representative if you add the energy determination in the samples. If you have this samples in your lab, I strongly recommend you determine the energy to include in this paper. Amino acid could be a very important analysis, but I understand it too expensive, but it is really a great information.

We agree that add data about energy  or amino acid will be extra information about digestibility, however we don’t have this samples that’s why we show only digestibility of DM, CP and EE.

Line 128 change “where” to “in which” or simply “and”

The relevant correction has been made L165

Lines 131 and 132 – “At 6 and 15 weeks of age, sixty birds (3 from each replicate) were slaughtered to analyze GIT structure and function”, are the same birds you used for digesta collection ? Is there no interference on this procedure of collection ?

Yes, there were the same birds, in which we analyzed GIT structure and collected digesta. It’s our standard procedure to take measurements and collect digesta simultaneously.

Lines 135 to 137 – “In digesta of the crop, gizzard, proventriculus, small intestine and ceca, the pH value was determined using a pH meter Small intestinal contents were collected for viscosity evaluation.” I believe there are two phrases here. Correct the punctuation, please.

The relevant correction has been made L 173

Lines 146 and 147: “The concentrations of short-chain fatty acids (SCFAs) were analyzed by gas chromatography (Shimadzu GC 14A, Shimadzu Co., Kyoto, Japan).” In which fractions of the GIT ?

The missing information was added. L184

Lines 148 to 150. “In weeks 6 and 15 of the experiment, excreta samples were collected from 10 turkeys in each group for microbiota analyses, which were performed immediately after sampling.” Why did you not collect cecal content ?

Many times digesta in ceca is very scarce. We didn't have enough samples to make a many analyzes, so we decided to analyze SCFA.

RESULTS

In the result description, please be more precise. The use of symbols or abbreviations make more difficult the comprehension of the text. You use BA and Sb, CSB, BAG all the time.

The relevant correction has been made through the manuscript.

In the first phrase “The final BW of birds fed BA-supplemented diets were similar to those of control group animals (Table 2). Birds receiving diets with the protected forms of BA were characterized by lower values of the FCR (p≤0.05) as compared with the control group. Turkeys whose diets were administered with protected forms of BA had higher values of the EEI (p≤0.01) than animals from the other groups.” It is not clear. Describe which treatment is different and compare to the active principles of the product. But use the correct description of the treatment to be more precise.

The relevant correction has been made L 209

The tables 4 and 5 need to be adjusted to be correctly formatted to the page. Please, do it to clear the viewing tables in the pages

The relevant correction has been made in table 4 and 5

DISCUSSION

The discussion is strongly based in literature comparison, but the real discussion must be improved. The problem is that the authors tested some variables that could not be compared to other papers, like digestibility and microbiome. It is possible to note that some bacteria genera determination cannot be extrapolated to a 16-S microbiome determination.

Other phrase that was not in accordance is the last paragraph in which the authors use non-statistical differences observed in body weight to explain the results.

I recommend that the discussion should be re-written to be closer to the variables the authors tested in order to explain better the results.

The discussion was rewritten.

CONCLUSIONS

The lines from 330 to 337 are repetition of the results. There is no conclusion, even a recommendation. Just a general information:  “The protected forms of BA can be valuable feed additives in turkey nutrition."

The conclusions was rewritten.

Reviewer 5 Report

Dear authors, this study has many problems which must be considered before gets published in Animals journal. First of all, the authors claim that there are no studies investigating the effects of butyric acid supplementation in poultry. However, with quick research many studies have been found to investigate the effects of butyric acid supplementation in broilers (Kazmarek et al 2016; Chaudari et al. 2020), laying hens (Jahanian and Golskali. 2015), and turkeys (Makowfski et al 2022-the same authors one month ago) diets on performance nutrient digestibility, gut morphology, and microbiota. In the material and methods, section authors did not refer according to which international ethic committee the turkeys of the present experiment were euthanized. Moreover, the statistical analyses is very weak, using Duncan tests instead of Tukey HSD post hoc tests and not using non-parametric tests for checking the normality of values. In addition, authors use the same results for the effects of butyric acid on turkeys' performance, with their publication in the same journal one month ago “The Effects of Different Forms of Butyric Acid on the Performance of Turkeys, Carcass Quality, Incidence of Footpad Dermatitis and Economic Efficiency” Zbigniew Makowski, Krzysztof Lipinski and Magdalena Mazur-Kusnirek, (Animals 2022, 12, 1458. https:// doi.org/10.3390/ani12111458). Finally, the authors need more biochemical and molecular analyses to exert the conclusion that Butyric acid which is used in the present study can be used as feed additive in turkey nutrition.

Due to these facts, authors must do better research and add more biochemical and maybe molecular analyses for this experiment plus the statistical analyses must be major revised to be published in Animals Journal. Conclusively, the present study must be rejected, and the authors must do many changes and additions in order to be published in the present journal.

Author Response

Dear authors, this study has many problems which must be considered before gets published in Animals journal. First of all, the authors claim that there are no studies investigating the effects of butyric acid supplementation in poultry. However, with quick research many studies have been found to investigate the effects of butyric acid supplementation in broilers (Kazmarek et al 2016; Chaudari et al. 2020), laying hens (Jahanian and Golskali. 2015), and turkeys (Makowfski et al 2022-the same authors one month ago) diets on performance nutrient digestibility, gut morphology, and microbiota.

According to the literature, there are scarce information about using different forms of butyric acid in turkey nutrition. In fact, in last month our article was published but we analyzed different indicators.

 In publication “ Makowski, Z., Lipiński, K. and Mazur-Kuśnirek, M., 2022. The Effects of Different Forms of Butyric Acid on the Performance of Turkeys The aim of the study was to compare the efficacy of butyric acid glycerides sodium butyrate and coated sodium butyrate in turkey nutrition based on the growth performance of birds, carcass yield, meat quality, the incidence of footpad dermatitis (FPD) and economic efficiency. In present study we try to determine the efficacy of different form of butyric acid in turkey nutrition based on an analysis of nutrient digestibility, gastrointestinal function and fecal microflora populations.

In the material and methods, section authors did not refer according to which international ethic committee the turkeys of the present experiment were euthanized.

According to the Journal style the statement about ethics committee is in the end of manuscript in paragraph "Institutional Review Board Statement".

Moreover, the statistical analyses is very weak, using Duncan tests instead of Tukey HSD post hoc tests and not using non-parametric tests for checking the normality of values.

In this research, where was multi stages and there was equal number of each group, we decided to use a one-way ANOVA analysis with post-hoc test (Duncan) to confirm where the differences occurred between a few groups.

In addition, authors use the same results for the effects of butyric acid on turkeys' performance, with their publication in the same journal one month ago “The Effects of Different Forms of Butyric Acid on the Performance of Turkeys, Carcass Quality, Incidence of Footpad Dermatitis and Economic Efficiency” Zbigniew Makowski, Krzysztof Lipinski and Magdalena Mazur-Kusnirek, (Animals 2022, 12, 1458. https:// doi.org/10.3390/ani12111458).

We think that show information about performance is a very important and interacts on results of whole work. In this publication “The effects of different forms of butyric acid on the performance of turkeys, carcass quality, incidence of footpad dermatitis and economic efficiency” was presented detailed results. In present study we give only overall bird’s growth performance.

 Finally, the authors need more biochemical and molecular analyses to exert the conclusion that Butyric acid which is used in the present study can be used as feed additive in turkey nutrition.

We agree that the more biochemical analyses would be useful to the main final conclusion, From the organization and economical point of view we could presented only this results. For sure, in our next trial we’ll think about molecular and biochemical analysis.

Due to these facts, authors must do better research and add more biochemical and maybe molecular analyses for this experiment plus the statistical analyses must be major revised to be published in Animals Journal. Conclusively, the present study must be rejected, and the authors must do many changes and additions in order to be published in the present journal.

Round 2

Reviewer 5 Report

Dear authors,

I believe that the necessary changes have been done. I propose that this work can be published in the Animals journal.